# H3K27me3 Depletion during Differentiation Promotes Myogenic Transcription in Porcine Satellite Cells

**DOI:** 10.3390/genes10030231

**Published:** 2019-03-19

**Authors:** Sheng Wang, Yan Sun, Ruimin Ren, Junhui Xie, Xiaohuan Tian, Shuhong Zhao, Xinyun Li, Jianhua Cao

**Affiliations:** 1Key Laboratory of Agricultural Animal Genetics, Breeding and Reproduction (Huazhong Agricultural University), Ministry of Education, Wuhan 430070, China; wangsheng@webmail.hzau.edu.cn (S.W.); Ysun@webmail.hzau.edu.cn (Y.S.); ruimin.ren@webmail.hzau.edu.cn (R.R.); jhxie@webmail.hzau.edu.cn (J.X.); xhtian@webmail.hzau.edu.cn (X.T.); shzhao@mail.hzau.edu.cn (S.Z.); 2The Cooperative Innovation Center for Sustainable Pig Production—Swine Breeding and Reproduction Innovation Platform, Huazhong Agricultural University, Wuhan 430070, China

**Keywords:** pig, H3K27me3, porcine satellite cells, differentiation, skeletal muscle

## Abstract

Background: Porcine skeletal muscle satellite cells play important roles in myogenesis and muscle regeneration. Integrated analysis of transcriptome and histone modifications would reveal epigenomic roles in promoting myogenic differentiation in swine. Methods: Porcine satellite cells (PSCs) were isolated and in-vitro cultured from newborn piglets. RNA Sequencing (RNA-Seq) and Chromatin Immunoprecipitation Sequencing (ChIP-Seq) experiments were performed using proliferating cells and terminal myotubes in order to interrogate the transcriptomic profiles, as well as the distribution of histone markers—H3K4me3, H3K27me3, and H3K27ac—and RNA polymerase II. Results: The study identified 917 differentially expressed genes during cell differentiation. The landscape of epigenetic marks was displayed on a genome-wide scale, which had globally shrunken. H3K27me3 reinforcement participated in obstructing the transcription of proliferation-related genes, while its depletion was closely related to the up-regulation of myogenic genes. Furthermore, the degree of H3K27me3 modification was dramatically reduced by 50%, and 139 myogenic genes were upregulated to promote cell differentiation. Conclusions: The depletion of H3K27me3 was shown to promote porcine satellite cell differentiation through upregulating the transcription level of myogenic genes. Our findings in this study provide new insights of the epigenomic mechanisms occurring during myogenic differentiation, and shed light on chromatin states and the dynamics underlying myogenesis.

## 1. Introduction

Porcine skeletal muscle is an important source of protein, and a model for studying human muscle-related diseases [1,2]. The development and regeneration of skeletal muscle mainly depends on the proliferation and differentiation of porcine skeletal muscle satellite cells [3,4]. As an adult muscle stem cell, porcine satellite cells (PSCs) usually exist in a resting state between the sarcolemma and the basal lamina of the muscle tissue, and play an important role in maintaining its normal physiological function [4,5]. The proliferation and differentiation of PSCs are regulated by a variety of factors, including histone modification, transcription factors, and long non-coding RNAs (lncRNAs) [6,7,8]. How histone modifications control and affect the proliferation and differentiation of PSCs remains largely unknown.

Previous studies have shown that the differentiation ability of satellite cells was distinct in different species. The differentiation efficiency of PSCs was less than 65%, and the process was regulated by various myogenic factors such as MYOD1, MYF5, MYOG, and MYF6 [7,9,10]. Second, epigenetic modifications like histone modifications are closely related to the transcriptional regulation of myogenic genes [3,11]. In mouse C2C12 cells, H3K27me3 histone modification regulates myogenic differentiation of satellite cells by regulating the expression of myogenic transcription factors, such as Myog, while H3K4me3 at the transcription initiation sites of *Myf5* and *MyoD* would respectively promote the proliferation and differentiation in C2C12 cells [12,13,14]. In addition, chromatin conformation rearrangements are also required for differentiation and muscle fiber formation. The polycomb protein Ezh2 mediates H3K27me3 to maintain the inhibitory chromatin conformation of myogenic genes, and to inhibit differentiation [15]. The methyltransferase *UTX* knockout mice showed that abnormal H3K27me3 modification leads to disordered muscle fiber regeneration [16]. In addition, decrease in H4K20me2 marks in *Suv-20h1* gene knockout mice resulted in genome-wide H3K27me3 depletion, thereby activating the resting satellite cells to commit myogenic differentiation [3]. However, whether the differentiation of PSCs is associated with histone modifications, especially the H3K27me3, is not well established.

In this study, PSCs were isolated from newborn piglets and cultured. Transcription and histone modifications were identified by genome-wide profiling of transcriptome and chromatin states, respectively. Our results identified 917 differentially expressed genes (DEGs), which were closely related to H3K4me3, H3K27ac, and H3K27me3. Furthermore, the genome-wide histone modification level significantly decreased during differentiation. Further, H3K27me3 was reduced by 50%, which largely led to the upregulation of 139 myogenic DEGs during PSC differentiation. The results of this study will provide clues for further studies on the relationship between myogenic DEGs and histone modifications, especially H3K27me3 underlying differentiation mechanism of PSCs, to further supplement the explanation for chromatin state and dynamics during skeletal muscle myogenesis in pigs.

## 2. Materials and Methods

### 2.1. Isolation, Culture, and Differentiation of Porcine Satellite Cells (PSCs)

Satellite cells were primarily isolated from hind leg muscles of one-week-old Yorkshire male piglets. Piglets were slaughtered according to a standard procedure approved by guidelines from the Regulation of the Standing Committee of Hubei People’s Congress (Hubei Province, China, HZAUSW-2017-008). Skeletal muscles were minced into pieces and digested in 0.2% type I collagenase (Sigma, USA, V900891) in a shaking water bath at 37 °C for 2 h. The supernatant cell suspension was washed with DMEM high glucose medium (Life, USA, 10569) supplemented with 1% antibiotic-antimycotic (Life, 15240) and 50 μg/mL gentamycin (Life, 15750), and sequentially passed through 100 μm, 70 μm, and 40 μm filters (BD, USA, 352360, 352350, 352340) to remove tissue debris. The cells were resuspended in RPMI-1640 medium (Life, A10491) supplemented with 20% FBS (Gibco, USA, 10099-141), 1% non-essential amino acids (Gibco, 11140-050), 0.5% chicken embryo extract (GEMINI, USA, 100-163P), 1% GlutaMax (Gibco, 35050), 1% Antibiotic-Antimycotic, 50 μg/mL Gentamycin, and 2.5 ng/mL bFGF (Life, 13256). The mixed cells were cultured in uncoated plates for 2 h to remove fibroblasts using differential adhesion property. The purified satellite cells were transferred into the Matrigel (BD, 356234) coated plates for proliferation cultures. At ~60% confluence, the proliferation medium was replaced by the DMEM high glucose medium (Life, 10569) supplemented with 5% horse serum (HyClone, USA, SH30074.02), 1% antibiotic-antimycotic, and 50 μg/mL gentamycin to induce PSCs from proliferation to differentiation state. Three time points of differentiation, 1-day (D1), 2-day (D2), and 4-day (D4) were chosen to investigate the differentiation efficiency on three replicates from 3 independent piglets.

### 2.2. Immunofluorescence Assay of PSCs

Cells were seeded in Matrigel coated 6-well plates overnight to approximately 60% confluence, and washed with cold PBS (HyClone, SH30256.01) twice. The cells were then fixed in 4% paraformaldehyde for 15 min, and incubated in 0.25% Triton X-100 (Amresco, USA, 0694) for 15 min at room temperature. The primary monoclonal antibodies, PAX7 (ABclonal, China, A7335), MYOD1 (ABclonal, A0671), DES (Abcam, UK, ab8976), MYOSIN (Abcam, ab15), and MYOG (Abcam, ab1835) were hatched with the prepared cells at 4 °C overnight. The secondary antibodies, anti-mouse IgG alexa fluor 555 (CST, USA, 4409s) and anti-rabbit IgG alexa fluor 488 (CST, 4412s), were incubated with the cells for 1 h in dark room. Cell nuclei were stained with Hoechst33342 (Sigma, B2261). Images were captured using the OLYMPUS IX73 TH4-200 system (OLYMPUS, Japan).

### 2.3. Validation of Differentially Expressed Genes (DEGs) by Quantitative PCR (qPCR)

Total RNA was extracted using RNeasy Mini kit (QIAGEN, Germany, 74104) and DNase I (QIAGEN, 79254). Reverse transcription was performed using PrimeScript RT reagent kit with gDNA Eraser (Takara, Japan, RR047A). Random primers and oligo-dT primer were added to initiate cDNA first strand synthesis. The quantitative PCR reactions were carried out using the SYBR Green master mix (TOYOBO, Japan, QPK-201) on a LightCycler 480 II (Roche, Switzerland) system. Data were calculated by the Delta-Delta-Ct algorism, using GAPDH as internal calibration on 3 replicate samples. All primer sequences are listed in the supplementary data (Appendix A).

### 2.4. Chromatin Immunoprecipitation (ChIP) and ChIP-Seq Library Preparation

Cells (10^7^) were cross-linked with 1% formaldehyde (EMD Millipore, USA, 344198) at room temperature for 10 min. The chromatin was obtained by cell lysis (50 mM HEPES, pH 7.5; 150 mM NaCl; 1mM EDTA; 1% Triton X-100; 0.1% sodium deoxycholate; 0.1% SDS; and proteinase inhibitors) and was followed by sonication (Sonics VCX130, USA) on ice for 6 min. ChIP-grade antibodies (10 μg), anti-H3K44me3 (Abcam, ab8580), anti-H3K27ac (Abcam, ab4729), anti-H3K27me3 (Millipore, 07-499), and anti-RNA polymerase II (Abcam, ab817), were incubated with 100 μL Protein G DynaBeads (Life, 10009D) at 4 °C for 2 h. The chromatin and the antibody coated beads were co-immunoprecipitated on a rotator at 4 °C overnight. The ChIP DNA was de-crosslinked by protein K (Ambion, AM2546) digestion at 55 °C overnight. DNA (1 μg) was used for the preparation of the Chromatin Immunoprecipitation Sequencing (ChIP-Seq) library by NEBNext UltraⅡ DNA Library Prep Kit for Illumina (NEB, USA, E7645 and E7335), according to the manufacturer’s instructions. Each ChIP-Seq library was high-throughput sequenced, ~50M paired-end reads (2 × 150 bp) on Illumina HiSeq3000 platform.

### 2.5. Library Construction and Sequencing

For each sample, total RNA was extracted and sent to Shanghai Biotechnology Corporation (Shanghai, China) for library construction. The RNA Sequencing (RNA-Seq) library of each sample was prepared by the NEBNext@ Ultra^TM^ II DNA Library Prep Kit (Illumina). And the library was sequenced paired-end reads (2 × 150 bp) on Illumina HiSeq3000 platform.

### 2.6. Data Preprocessing and Alignment

The sequencing raw data was first filtered to remove low-quality reads by fastQC, and adapters were removed using Cutadapt. The clean reads were then mapped to the pig genome (Ensembl Sscrofa 11.1) by Bowtie2 with default parameters [17]. The unique mapping reads were further processed to remove duplications based on genomic coordinates. The reproducibility of de-duplication reads was analyzed by calculating the coefficients of two samples. Briefly, the genome is first divided into a number of 10 kb bins, then the reads of two samples (de-duplication unique mapping bam file) on each bin are taken as the coordinates of the x-axis and y-axis, respectively. Draw the reads on all bins in the form of a scatter plot and calculate the Pearson correlation coefficient. The closer reads number in a bin are to the samples, the more accumulation points are on the diagonal, the greater the correlation coefficient, indicating that the better repeatability of two samples [18].

### 2.7. RNA-Seq Data Analysis

HTSeq software was used to count the reads for RNA-Seq data, after pre-processing. Further data mining was subsequently conducted including: DEGs analysis by DESeq2 package with default parameters, gene ontology (GO) analysis by DAVID and clusterProfiler package [19], further analysis was performed using GOplot with the results from DAVID [20], which are all available R packages [21].

### 2.8. ChIP-Seq Data Analysis

The ChIP-Seq peak calling was performed by MACS2 with a 3 kb extension of promoter regions. The parameters of regular mode were used for peak calling of H3K4me3 H3K27ac and RNA polymerase Ⅱ (RNAPII), which were ‘macs2 callpeak -t ChIP.bam -c Control.bam -f BAMPE -g 2.7e+9 -n name -B -q 0.05′. While broad mode was used as ‘macs2 callpeak -t ChIP.bam -c Control.bam -f BAMPE -g 2.7e+9 -n name --broad --broad-cutoff 0.05′ for peak calling of H3K27me3 [22]. The modifications of three histone marks (H3K4me3, H3K27ac, and H3K27me3) and RNAPII were normalized with input files as control using bamCompare of deepTools based on log2 fold change of the reads per kilobase million (RPKM) [23]. Principal component analysis (PCA) was used to compare the correlation of ChIP-Seq data with multiBigwigSummary and plotPCA of deepTools with normalized files using spearman model with default parameters, and the enrichment profiles were performed using computeMatrix, plotProfile and plotHeatmap of deepTools. The annotation of the modification regions of three histone modifications and RNAPII were illustrated by ChIPseeker ± 3 kb of the transcription start site (TSS) region [24].

### 2.9. Statistical Analysis

All results were presented as mean ± standard error of mean (SEM). Unpaired Student’s *t*-test was used to show the statistical significance. *p* <0.05 indicated the significant difference.

### 2.10. Data Availability

RNA-Seq and ChIP-Seq data were submitted to NCBI Sequence Read Archive database (SRP180031 and SRP180432).

## 3. Results

### 3.1. The Morphology Characterization of PSCs

Primary PSCs were isolated from skeletal muscle of newborn piglets. The proliferation (PRO) and differentiation (DIF) cells (D1, D2, D3 and D4) were illustrated by cell morphology. PSCs proliferated and renewed in PRO medium, and the significant myotube fusion was observed at D2 and D4 (Figure 1A). Furthermore, the PRO and D2 cells were characterized by PAX7 immunofluorescent cell (IFC) staining, which indicated the attribution of satellite cells. The transcription factors of differentiation, MYOD1 and MYOG, were also checked by IFC staining for PRO and D2 cells (Figure 1B). The PSCs used in this study were purified by pre-plating procedures. Based on PAX7 staining, more than 93% of positive PRO cells were positive, few however found in D2 cells (Appendix A). The PSCs possessed highly myogenic differentiation potentials, and specific differential markers—MYOG and MYOSIN—were only found in D2 cells that indicated myotube formation and myofibres establishment (Appendix A).

### 3.2. Characterization of Gene Transcription

The PRO and DIF (D2) cells were subject to gene transcriptional analysis using RNA-Seq, and comparative analysis of differentially expressed genes between the two profiles was performed. The results showed that there were 15,569 and 15,200 genes expressed in PRO and DIF, respectively (Appendix A). Moreover, 917 genes were identified as DEGs, which composed of 406 and 511 down- and up- regulated genes, respectively. The t up-regulated DEGs included *MYL2*, *TNNT1* and *MYL9,* indicating skeletal muscle properties, while the down-regulated DEGs such as *CDCA5*, were more involved in cycles related to cell proliferation (Figure 2A). The gene ontology (GO) analysis further demonstrated that down-regulated DEGs were significantly enriched in the kinetochore formation, cell cycles and spindle pole terms, which are related to cell proliferation. Meanwhile, the up-regulated DEGs were found related to Z disc structure, myofibril formation, and sarcomere terms, which are essential for physiological functions of skeletal muscle (Figure 2B). Furthermore, down tendentious dominated the GO terms in this study, only nine biological processes were demonstrated to be significantly upregulated by gene function analysis (Figure 2C). In order to elucidate the roles of DEGs in myogenesis, further analysis showed that 62 DEGs were linked to five top GOs related to myofibres functions. Interestingly, the Z disc and steroid biosynthesis terms only linked with up-regulated DEGs, such as *MYH7* and *SC5D*, regulating myofibres formation, while microtubule movement and kinetochore terms only connected with down-regulated DEGs, which participated in cell cycle modulation. Genes related to the *FoxO* signaling term, however, are involved in the whole myogenesis process of PSCs (Figure 2D). *TGFB3* was upregulated, while *CCNB1* and *CCNB3* were downregulated in promoting myogenic differentiation, which were differentially expressed during PSC differentiation, as shown by quantitative PCR (qPCR) (Appendix A).

### 3.3. Epigenome Alteration in the Differentiation of PSCs

To gain epigenomic insights into the mechanisms underlying the differentiation of PSCs, the chromatin stats alteration was surveyed by performing ChIP-Seq analysis of H3K4me3, H3K27ac, H3K27me3, and RNAPII (Figure 3A). The active promoter regions were mainly marked by H3K4me3, H3K27ac, and RNAPII and the distal repressed regions, however, were massively occupied by the H3K27me3 modifications, allowing us to interrogate the distributions of the epigenetic observations during cell differentiation. Principal component analysis (PCA) compared three histone modifications and RNAPII ChIP-Seq data of PRO and DIF cells, which were split into four parts explicating by function of the histone modification and cell phases. Active markers, H3K4me3, H3K27ac, and RNAPII, were separated from H3K27me3 modification. The cell phases were aggregated into PRO and DIF cell parts distinct from each other, suggesting the dynamic change of histone modifications of DIF from PRO cells (Figure 3B). Furthermore, the correlation was positive in PRO active and DIF active parts, between 0.28–0.45 using spearman model (Appendix A). Moreover, based on peak calling the genome wide modifications globally decreased during the differentiation of PSCs, especially H3K27me3, which showed remarkable depletion by 50% from 22,060 to 11,573 compared to PRO cells (Figure 3C). In order to investigate the relationship between transcriptional and epigenomic modifications, the integrated analysis showed that gene transcription was highly linked with RNAPII activation, H3K4me3, and H3K27ac modification. Interestingly, there were large numbers of proximal regions of genebody such as transcription end site (TES), which were also activated as indicated by RNAPII binding, but no gene transcription was accompanied by H3K4me3 modifications. The H3K27me3 modification, however, was only associated with chromatin repression states, so that shutting down gene transcription in those regions (Figure 3D). The region, chr15: 81,888,000–82,000,000 for instance, showed large fractions of H3K4me3, H3K27ac, and RNAPII peaks, which were enriched at the TSS of activated genes, and H3K27me3 peaks were captured at genebodies of silenced genes (Appendix A).

### 3.4. The Regulation Roles of Epigenetic Modification

To reveal the influence of epigenetic modifications on cell differentiation through gene expression regulation, epigenetic marks were interrogated across the 917 identified DEGs. The integral changes in RNAPII and H3K27ac were consistent with the trend of gene activation and repression (Figure 4A). H3K4me3 modification showed a downregulated trend, and only 124 genes were upregulated with increased H3K4me3 marks at the TSS regions (Appendix A). H3K27me3 modification, which was depleted at the TSS regions of upregulated genes, changed gently at a low level at the TSS and genebody regions of downregulated genes. Moreover, gene activation was demonstrated by H3K27ac modification with significant ascent at the TSS region of upregulated genes (Figure 4B). The DEGs necessary for muscle development, *MEF2C*, *MYBPH* and *CASQ2*, were described to show the influence of epigenetic modifications (Appendix A). *SHISA2* was highlighted for gene activation through the elevation of H3K27ac and RNAPII accompany with the depletion of H3K27me3 modification at the TSS region (Figure 4C).

### 3.5. The Roles of H3K27me3 During Cell Differentiation

To determine the specific roles of H3K27me3 histone modification during PSC differentiation, genome-wide screening was performed. Distinct reductions were found especially in intergenic, intronic, and promoter regions, accompanied by a significant decline in histone-modified genes (Figure 5A). Among the 917 identified DEGs, 179 were labeled with H3K27me3 histone modification (Appendix A), of which, 81 had H3K27me3 modifications only during the proliferation phase (Figure 5B). Various depletion of H3K27me3 was found at the TSS regions of 179 H3K27me3 marked genes, which upregulated 139 genes (Figure 5C). Moreover, the upregulated histone-modified DEGs were involved in skeletal muscle tissue development and negative regulation of cell proliferation terms, indicating the specific roles of H3K27me3 modification for gene activation during PSC differentiation (Appendix A). It was observed that the myogenic transcription factors and cell cycle related genes, such as *MYOG* and *CDKN2B*, were blocked with H3K27me3, of which the transcriptional activity was low at the proliferation phase (Figure 5D). Through H3K27me3 depletion, PSCs were promoted from the exiting cell cycle to initial cell differentiation with the up-regulation of *MYOG* together with cofactors, *MEF2C*, which would be released from repressive marks to activate downstream myogenic genes (Figure 5E).

## 4. Discussion

In this study, satellite cells of porcine skeletal muscle were isolated, primarily cultured, and differentiated into myotubes in vitro, presenting an advantage for research on myogenesis in pigs. Three histone modification marks, H3K4me3, H3K27ac, and H3K27me3, together with RNAPII, were interrogated in myoblasts and myotubes to illustrate the relationship between transcriptomic profiles and histone modifications during differentiation. This study provides evidences on H3K27me3 modification depletion in PSCs, which promote myogenic differentiation. The genome-wide integrated analysis that combined epigenomics and transcriptomics to illustrate the relationship between H3K27me3 and myogenesis will be helpful for better understanding myogenesis in pigs.

Primary PSCs were purified by pre-plating and fused into the myotubes within 48h of culture in differentiation medium, less than the 72 h previously reported [9]. Moreover, Transcriptomic analysis revealed that during cell differentiation a large number of muscle development-related genes were upregulated, in addition to metabolism-related genes including *SHISA2* and *FASN* [25,26]. Cell cycle-related genes were significantly downregulated promoting PSCs differentiation, which were consistent with previous studies [27]. Our results show that these important genes were differentially expressed during PSC differentiation, which were also confirmed by qPCR.

Intriguingly, previous studies mentioned that epigenetics played important roles in shifting heterochromatin to open chromatin for myogenic differentiation [28,29]. The dramatic depletion of H3K27me3 histone modification was observed during PSCs differentiation (~50%). Furthermore, 139 myogenic DEGs were upregulated through H3K27me3 depletion, which enriched not only the terms necessary for myotube formation such as muscle tissue development, but negative biological processes of cell proliferation and neural development pathways as well. The rapid reduction in H3K27me3 mark was also found during specific stages of embryogenesis and stem cell differentiation [13,30,31], as demonstrated by Dilworth using *UTX* conditional knockout mice, suggesting the necessity of H3K27me3 depletion [16,32]. In addition, it was believed that the dynamic balance between methyltransferase *EZH2,* demethyltransferase *JMJD3,* and *UTX* regulated the H3K27me3 depletion during cell differentiation [33,34]. Consistently, the expression of *EZH2* was shown to be significantly downregulated—barely detected in myotubes of PSCs. It was also found that the genes related to cell fate development, as well as maintenance and differentiation, were marked with downregulated H3K27me3 histone modifications, such as the gene regions of *HOXD* cluster regulating the biomorphic development.

Furthermore, molecular mechanisms have increasingly uncovered that histone modification promote myogenesis. Our results emphasized the dramatic decrease in H3K27me3 modifications during cell differentiation, and genome-wide mapping of three epigenetic marks together with RNAPII demonstrated that the myogenic transcription factors were repressed with H3K27me3 modifications during proliferation phase, such as MYOG, which would drive myogenesis through H3K27me3 depletion. While the reduction of H3K27me3 modification was also reported at the TSS region of *Myog* in C2C12, that lied at the key position in myogenic differentiation [13,16]. Consistent with previous results, the terminal differentiation genes, which were induced by *MYOD1* and *MYOG* such as *TNNC1* and *MYL9*, were modified with positive marks (H3K4me3, H3K27ac, and RNAPII) instead of H3K27me3 for promoting myotube formation [13]. In addition, significant myogenic characteristics were demonstrated in PSCs, which confirmed that myogenic transcription factors escaped from H3K27me3 histone modifications during cell differentiation, such as SIX4 and MYOD1 [35]. The critical transcription factors of adipogenesis, osteogenesis, and muscle inhibitory genes such as PPAR-γ and PRDM16, were labeled with Polycomb repression at the TSS region to show tissue specificity, which would convert cell identity [13,36,37]. Our result suggested the spatiotemporal specificity of H3K27me3 modification in releasing myogenic factors, which would activate downstream genes to regulate cell differentiation and myotube formation.

This study revealed that the significant depletion of H3K27me3 modification promotes PSCs differentiation. Our results extend the genome-wide understanding of the roles of H3K27me3 marks in regulating myogenic differentiation in pigs. However, further studies are required to fully dissect the epigenetic remodeling and epigenetic mechanism during PSCs differentiation.

## Figures and Tables

**Figure 1 genes-10-00231-f001:**
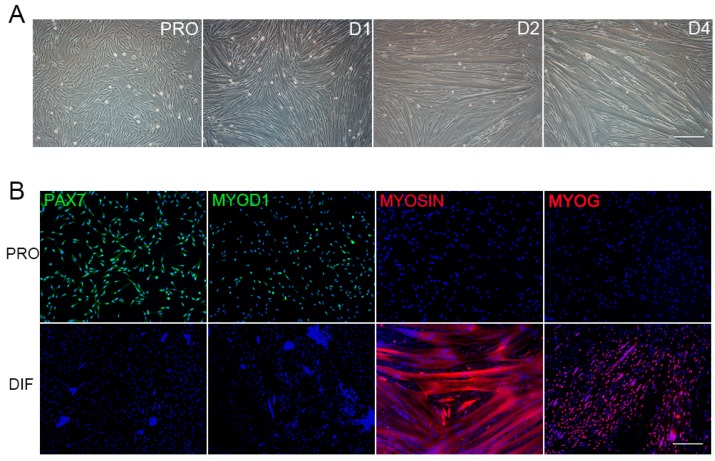
Porcine Satellite Cells (PSCs) culture, differentiation, and immunofluorescence detection. (**A**) The proliferation PSCs (PRO) were differentiated for 4 days in differentiation medium (D1, D2, and D4), and myotubes aroused at D2, with their size increasing over time. Scale bars: 200 μm. Magnification: 100×. (**B**) Immunofluorescence detection of four marker genes in proliferation cells (PRO, upper panel), and D2 differentiated cells (DIF, bottom panel). Antibodies were indicated as green (left to right: PAX7, MYOD1) and red (left to right: MYOSIN, MYOG), Hoechst33342 dye was blue. Scale bars: 200 μm. Magnification: 100×.

**Figure 2 genes-10-00231-f002:**
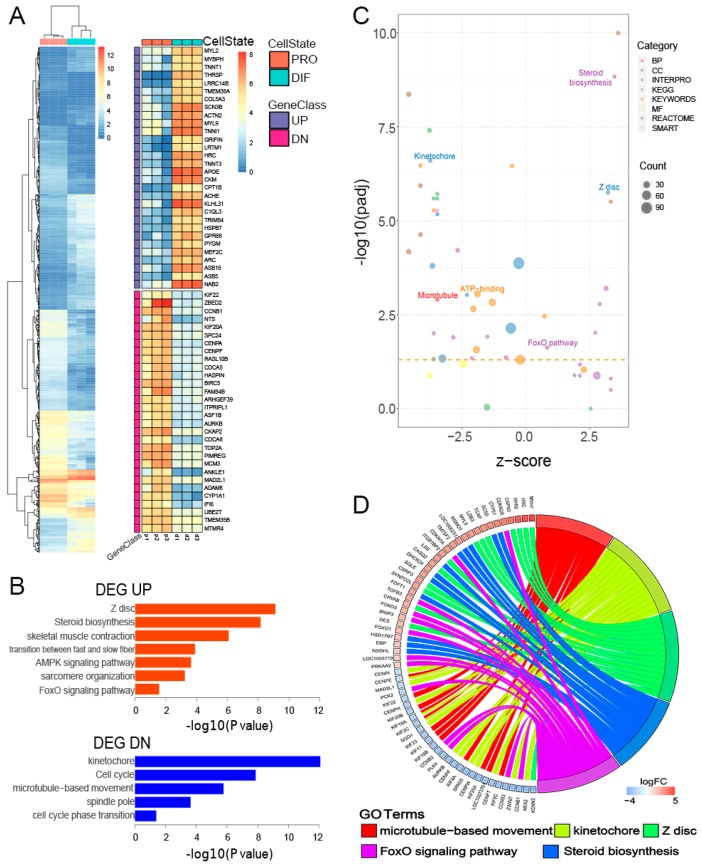
Transcriptomic profiles of PRO and DIF PSCs. (**A**) Heatmap of 917 differentially expressed genes (DEGs) interrogated between PRO and DIF cells. Sixty significant DEGs of up- and down-regulated genes illustrated the transcriptional patterns in two cell phases. (**B**) Gene ontology (GO) analysis of up- (UP) and down- (DN) regulated DEGs changed during cell differentiation. The enriched terms were ranked by -log10(*p* value). (**C**) The GO term enrichment was displayed, which was calculated using z-score for functional clusters, and six myogenesis-related GO terms were highlighted in the bubble plot. (**D**) The five most important pathways and biological processes were identified, and the DEGs were selected to reveal their function in muscle formation.

**Figure 3 genes-10-00231-f003:**
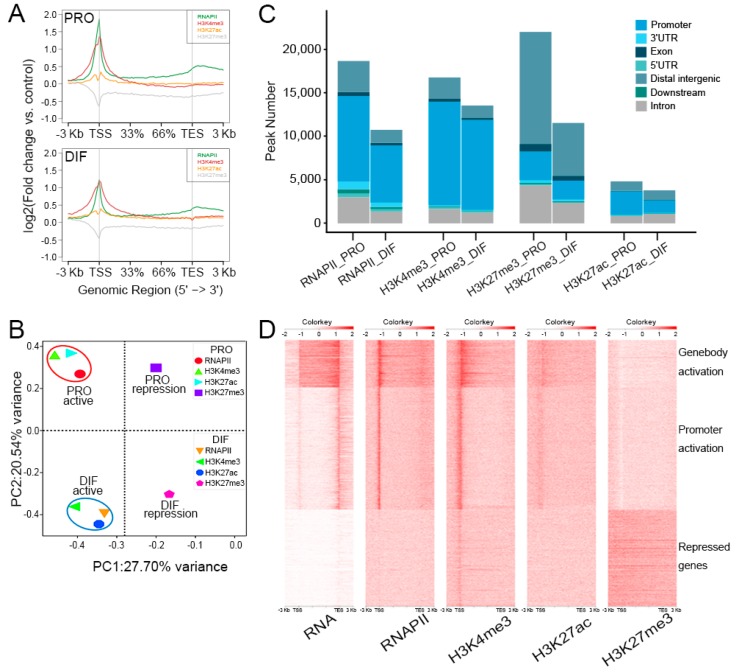
The insight of epigenomic alteration during PSCs differentiation. (**A**) Chromatin Immunoprecipitation Sequencing (ChIP-Seq) enrichment were plotted at −3 kb of the transcription start site (TSS) and +3 kb of the TES to show the presumable binding sites of three epigenetic marks and RNA polymerase Ⅱ (RNAPII) of PRO (upper panel) and DIF (bottom panel) cells. The y-axis was the average log2 fold change compared with control. (**B**) Data from three histone modifications and RNAPII between PRO and DIF cells were clustered into four parts according to cell phases and histone functions, which showed the distinction of histone modifications during cell differentiation. The first and second principal components were depicted as x-axis and y-axis. (**C**) The peak numbers of three epigenetic marks and RNAPII were described, and chromatin was illustrated into seven parts to show the distribution of modification peaks. (**D**) ChIP-Seq enrichments were plotted using a heatmap, which demonstrated the depositing of three histone modifications and RNAPII, and the correlation associated with gene expression profiles of myotubes. Maps were generated from 3 kb upstream of the TSSs to 3 kb downstream of the TESs. Three clusters were labeled according to gene expression and histone modification levels.

**Figure 4 genes-10-00231-f004:**
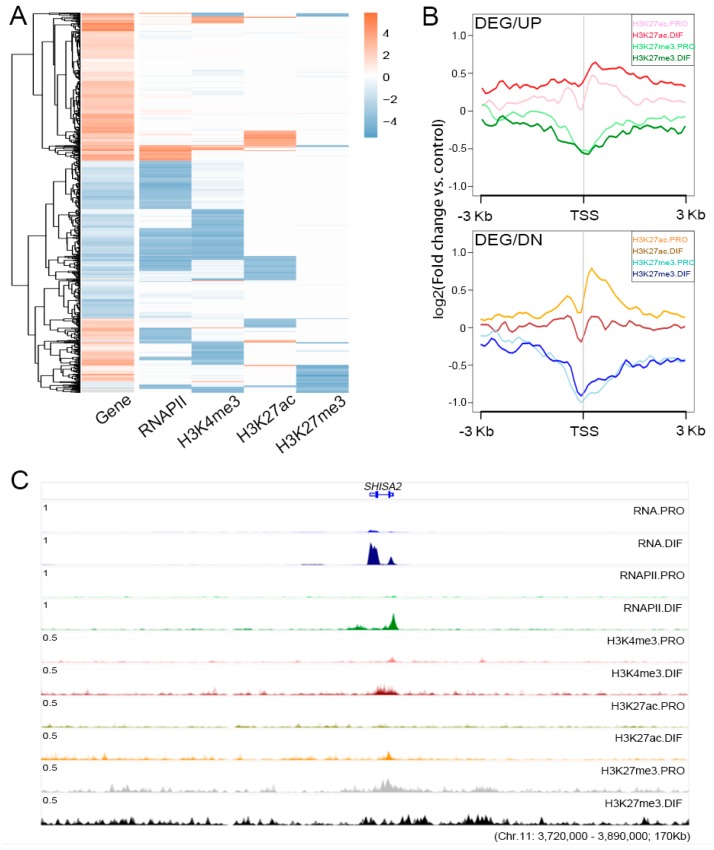
The relationships between epigenetic modifications and transcriptomic profiles during cell differentiation. (**A**) Heatmap illustrating the relationships between the changes of histone modifications and gene expression for the 917 DEGs during cell differentiation. (**B**) Expression profiles of H3K27ac and H3K27me3 were plotted at ±3 kb of TSS region of DEGs to show the presumable binding sites between PRO and DIF cells. Up (DEG/UP, upper panel) and down (DEG/DN, bottom panel) regulated DEGs were described separately. The y-axis was the average log2 fold change compared with control. (**C**) The integrative genomics viewer (IGV) browser was used to show ChIP-Seq and RNA Sequencing (RNA-Seq) enrichment for the up-regulated DEG, *SHISA2*, which showed that the gene was activated by the increase in RNAPII and H3K27ac combined with the depletion of H3K27me3 modification during cell differentiation.

**Figure 5 genes-10-00231-f005:**
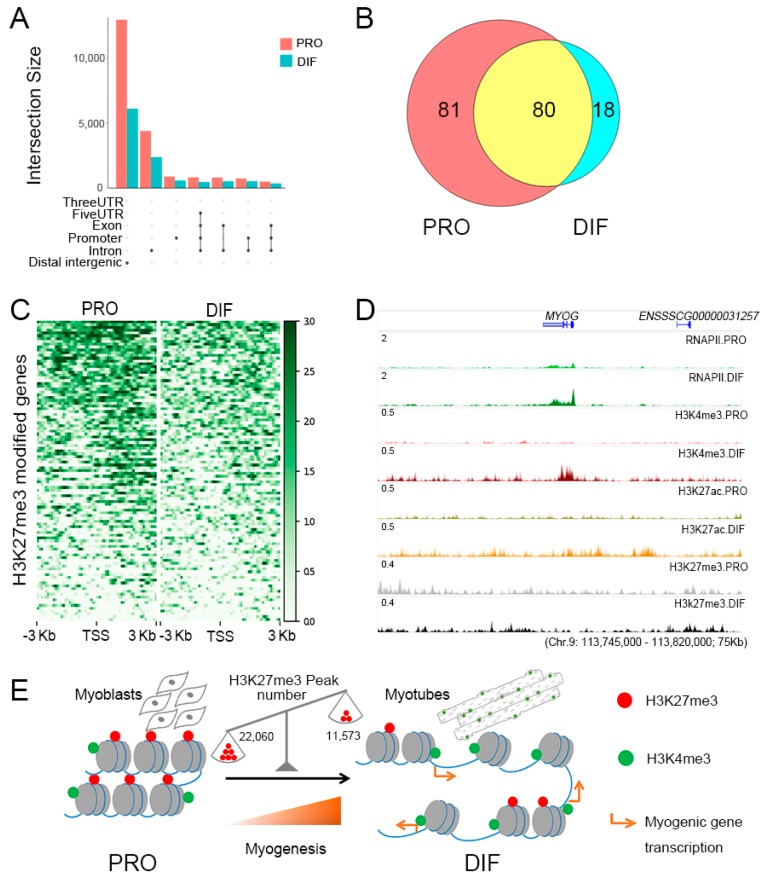
The role of H3K27me3 modification in regulating gene expression in PSCs. (**A**) The depletion regions of H3K27me3 modification are annotated with intersection size during cell differentiation. (**B**) Venn diagram of 179 H3K27me3 modified DEGs between PRO and DIF cells. (**C**) Heatmaps of H3K27me3 modification enrichment of 179 H3K27me3 modified DEGs between ±3 kb of TSS region of two cell phases, showing the depletion of H3K27me3 marks during cell differentiation. (**D**) The IGV browser was used to show ChIP-Seq and RNAPII enrichment for myogenic gene, *MYOG*, which was released from H3K27me3 histone modification during cell differentiation. (**E**) Simplified model of the roles of H3K27me3 depletion in promoting cell differentiation through upregulating myogenic genes. Red points presented H3K27me3 histone modification, which showed with the scales that the peak number of H3K27me3 modification was reduced from 22,060 to 11,573 during cell differentiation.

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
