# Peer review of "H3K27me3 Depletion during Differentiation Promotes Myogenic Transcription in Porcine Satellite Cells"

_genes, 2019, doi:10.3390/genes10030231_

Round 1
Reviewer 1 Report
The study by Wang et al., focused on the crucial roles of H3K27me3 modification during porcine satellite cell differentiation. Genome-wide mapping of three epigenetic marks H3K4me3, H3K27me3 and H3K27ac, together with transcription factor RNA polymerase II was employed. The depletion of H3K27me3 was demonstrated to promote porcine satellite cell differentiation and drive myogenesis through upregulating the transcription level of myogenic genes, such as MYOG. This is an interesting and well-designed study that sheds light into the epigenomic regulation of myogenic differentiation and would be of great interest to the audience of chromatin and gene regulation field. The manuscript is also well-written and nicely presented.
Author Response
Thank you for sparing your time reading this article. And it is my pleasure to achieve your approval.

Reviewer 2 Report
The authors performed genome wide signal profiling of three histone markers as well as pol II and RNA-seq of proliferated and differentiated porcine satellite cells. By Integrated analysis of this data they attempted to get insights into the role of epigenetic changes in promoting myogenic differentiation. Overall, this is an interesting and well-performed study. The analysis performed is intensive and thorough and the results are in agreement with what is known in the literature regarding the role of the tested histone modifications in gene regulation. Still I have few comments listed below.
Major comment:
1. No biological replicates of the ChIP-seq were performed (or described) in the study and it is not possible to determine to what extent the observations are reproducible.
Other comments:
Introduction
2. Line 66: what do you mean by “closely related”? the term is not clear, it can be that histone modifications regulate DEGs expression, or literally that these modifications located in close proximity to DEGs. Please explain this term.
3. Line 67: “H3K27me3 degree was reduced by 50%, which largely upregulated 139 myogenic DEGs, thereby promoted the differentiation from satellite cells into myotubes.” - There is no direct evidence in the paper for the role of H3K27me3 in promoting differentiation. The data presented is mainly correlative. This can be tested by knock down of the enzyme required for H3K27me3 demethylation to address whether its loss influences differentiation capacity of the cells. The authors should avoid jumping into conclusions not supported by the data.
Methods
4. Description of RNA-seq preparation and RNA-seq data analysis are missing from the method section
5. Line 129: “The reproducibility of de-duplication reads was analyzed by calculating the coefficients of two samples.” – not clear, please elaborate
6. Line 131: “MACS2 with 3 kb extension of promoter regions” – it’s not clear what the author mean by extension of promoter regions. MACS extends reads to in 5'->3' direction but this is done to all reads not only in promoter regions. Please specify the parameters used in MACS2 so one could reproduce the analysis. In addition, some of the tested modifications (such as H3K27me3) have diffuse signal, please specify which parameters were used to detect broad peaks.
Results
7. Statistics regarding number of sequenced reads and mapping should be added to supplementary information.
8. Line 172: “The results showed that there were 155,569 and 15,200 genes detected as transcription in PRO and DIF” – I suppose there is a mistake in the number 155,569 (the pig genome contains ~21000 protein coding genes). Also the term “detected as transcription” is not clear, use “expressed” instead.
9. In figure 2A there are two heatmaps, one of 917 DEGs, and the other of 60 selected DEGs. Please explain why you selected these 60 (out of the 917).
10. How figure 2C+D generated? If you used a specific package for this (it looks like GOplot R package) then please add the relevant citation and details in the method section.
11. Quality of figure2 (and other figures as well) is very low. Also, please replace the GO terms in 2C with terms description, which is much more informative, or add a legend next to the figure with the GO terms description.
12. Line 184 – “chord analysis” please change this term. It will not be clear to readers who did not perform GOplot analysis.
13. Figure S2D – please add p values
14. Line 210: “negative correlation was showed between H3K27me3 and other histone modifications” – I can’t see this in figure S3A. Based on the figure the correlation between H3K27me3 and other modifications is positive. It is also not clear how the correlation was calculated. A description should be added to the method section. If it based on consecutive sized bins across the whole genome then mostly background signal from the genome is taken into account. It would be more informative to calculate the correlation based on enriched regions.
15. Which analyses were performed on peak regions? and which on genome wide signal? More details should be added.
16. Figure 4A –Is the color scheme relate to both genes and modifications? How the histone signal was summarised across DEGs in order to generate this plot? gene bodies? promoters?
17. Please add Y axis label to figure S4A, and correct the legend (change the second (A) to (B)
18. Figure 5A is hard to interpret, it will be more informative to present the distribution of H3K27me3 peaks across genomic features using pie charts instead.
19. Line 268: “Moreover, the upregulated 268 histone-modified DEGs were remarkable enriched in skeletal muscle tissue…” – The number of genes represented within these processes is between 2-4 and the enrichment is not highly significant. Taking this into consideration please use soft term instead of “remarkable enriched”, and also be careful in drawing conclusions regarding the role of H3K27me3 modification for gene activation during PSC differentiation from this observation.
20. Line 273 – “MYOG and CDKN2B, were blocked with H3K27me3, of which the transcriptional activity was low at proliferation phase (Figure 5D) “ – while the figure definitely show increase in MYOG H3K4me3 labeling in DIF cells, no reduction in H3K27me3 is clearly seen. The authors should update the text and figure legend to describe observations that are supported by the data. Also please correct the legend “The IGV browser was used to show the ChIP-Seq and RNA-Seq enrichment” it should be “RNA pol II” instead of “RNA-seq”
21. List of DEGs should be given as supplementary table
22. Line 299: “Our results showed that it is the differential expression of these DEGs initiating cell differentiation to form myotubes, which were confirmed by qPCR.” – no experiments were done to test that the change in expression of the DEGs initiate cell differentiation. What the authors show is that these genes change their expression in differentiating cells. Please rephrase.
23. Line 318: “Our results emphasized the crucial roles of H3K27me3 modification during cell differentiation” – again, the results show that H3K27me3 modification dynamically changed during differentiation. In order to emphasize the crucial roles of H3K27me3 for differentiation, experiments should be done. Please rephrase.
24. English editing is required, please also change "multi-comb" to "polycomb" in line 56
Author Response
Reviewer 2:
The authors performed genome wide signal profiling of three histone markers as well as pol II and RNA-seq of proliferated and differentiated porcine satellite cells. By Integrated analysis of this data they attempted to get insights into the role of epigenetic changes in promoting myogenic differentiation. Overall, this is an interesting and well-performed study. The analysis performed is intensive and thorough and the results are in agreement with what is known in the literature regarding the role of the tested histone modifications in gene regulation. Still I have few comments listed below.
Major comment:
1. No biological replicates of the ChIP-seq were performed (or described) in the study and it is not possible to determine to what extent the observations are reproducible.
Response 1: The reviewer’s points just hit in the bull's eye, and we also agree that it should has biological replicates in this study. The main reason for this problem is the lack of experience (also funds) in this area. Despite this flaw in the data, we can still speculate that its reproducibility is good from these data. We additionally analyzed the H3K4me3 signal and reproducibility of chromosome 1 (the largest chromosome in pigs) and found that: (1) the signal-to-noise ratio is very good, and the cross-correlation shows that the DIF and PRO have a highly consistent position and length of phantompeak (~240bp) (FigureS6A); (2) We also compared the peak reproducibility between these two states. The results are also highly correlated (r = 0.97) (FigureS6B), indicating that most of the H3K4me3 modification locations remain unchanged, which is consistent the literature and the actual biological situation. Therefore, we believe that the high quality of data can reduce the probability of false positive discovery to a certain extent, but unfortunately it is difficult to repeat the whole experiment in the current situation.
Introduction
2. Line 66: what do you mean by “closely related”? the term is not clear, it can be that histone modifications regulate DEGs expression, or literally that these modifications located in close proximity to DEGs. Please explain this term.
Response 2: About the term “closely related” at line 66, it was used to illustrate that histone modifications regulated the expression of these DEGs.
3. Line 67: “H3K27me3 degree was reduced by 50%, which largely upregulated 139 myogenic DEGs, thereby promoted the differentiation from satellite cells into myotubes.” - There is no direct evidence in the paper for the role of H3K27me3 in promoting differentiation. The data presented is mainly correlative. This can be tested by knock down of the enzyme required for H3K27me3 demethylation to address whether its loss influences differentiation capacity of the cells. The authors should avoid jumping into conclusions not supported by the data.
Response 3: Indeed, as reviewer said, the conclusion was not mainly correlative, and the sentence would change into “Further, the H3K27me3 degree was reduced by 50%, which largely upregulated 139 myogenic DEGs during PSC differentiation.”
Methods
4. Description of RNA-seq preparation and RNA-seq data analysis are missing from the method section
Response 4: Added the description of RNA-seq preparation and RNA-seq data analysis in “Materials and Methods” section.
5. Line 129: “The reproducibility of de-duplication reads was analyzed by calculating the coefficients of two samples.” – not clear, please elaborate
Response 5: According to the reviewer's request, the brief explanation has been added here and the relevant literature of the method has been added.
6. Line 131: “MACS2 with 3 kb extension of promoter regions” – it’s not clear what the author mean by extension of promoter regions. MACS extends reads to in 5'->3' direction but this is done to all reads not only in promoter regions. Please specify the parameters used in MACS2 so one could reproduce the analysis. In addition, some of the tested modifications (such as H3K27me3) have diffuse signal, please specify which parameters were used to detect broad peaks.
Response 6: About the term “MACS2 with 3 kb extension of promoter regions” at line 131, it was regarded as 3 kb upstream of the TSSs to 3 kb downstream of the TESs to generate the results of peaks according to previous studies (Asp, P. et al., 2011; Liu, L. et al., 2013). Although 2 kb and 3 kb were all mentioned previously, 3 kb were selected in this study to generate more information in our results. And the parameters of MACS2 were added in ChIP-seq data analysis of “Materials and Methods” section, which mentioned that broad mode was only used in H3K27me3 modification peak calling.
Results
7. Statistics regarding number of sequenced reads and mapping should be added to supplementary information.
Response 7: The data of number of sequenced reads and mapping were added into table S1 of supplementary.
8. Line 172: “The results showed that there were 155,569 and 15,200 genes detected as transcription in PRO and DIF” – I suppose there is a mistake in the number 155,569 (the pig genome contains ~21000 protein coding genes). Also the term “detected as transcription” is not clear, use “expressed” instead.
Response 8: The number of the detected genes was corrected from “155,569” to “15,569”. The term “detected as transcription” would be replaced and the sentence was changed into “The results showed that there were 15,569 and 15,200 genes expressed in PRO and DIF”.
9. In figure 2A there are two heatmaps, one of 917 DEGs, and the other of 60 selected DEGs. Please explain why you selected these 60 (out of the 917).
Response 9: The reason of choosing these 60 DEGs was based on significantly value of fold change and p value between proliferation and differentiation cell phases. The top 30 DEGs of the most upregulated and downregulated DEGs was selected respectively, which screened the novel genes.
10. How figure 2C+D generated? If you used a specific package for this (it looks like GOplot R package) then please add the relevant citation and details in the method section.
Response 10: Figure 2C was generated by GOplot R package using the gene ontology results from DAVID, and Figure 2D was also generated by GOplot with 62 selected DEGs which were linked with 5 top GO terms necessarily in myogenesis. Relevant citation and details of these two figures were added in “Materials and Methods” section.
11. Quality of figure2 (and other figures as well) is very low. Also, please replace the GO terms in 2C with terms description, which is much more informative, or add a legend next to the figure with the GO terms description.
Response 11: The results of Figure2 were updated to increase resolution. Relevant description was highlight in Figure 2C to annotate 6 myogenesis related GO terms, and the legend was modified in Figure 2C.
12. Line 184 – “chord analysis” please change this term. It will not be clear to readers who did not perform GOplot analysis.
Response 12: The term “chord analysis” at line 184 would be replaced. And the sentence was changed into “further analysis showed that 62 DEGs were linked with 5 top GOs related with myofibres functions.”
13. Figure S2D – please add p values
Response 13: Added the p values for each result in supplementary Figure S2D.
14. Line 210: “negative correlation was showed between H3K27me3 and other histone modifications” – I can’t see this in figure S3A. Based on the figure the correlation between H3K27me3 and other modifications is positive. It is also not clear how the correlation was calculated. A description should be added to the method section. If it based on consecutive sized bins across the whole genome then mostly background signal from the genome is taken into account. It would be more informative to calculate the correlation based on enriched regions.
Response 14: The sentence would delete as faulty description was mentioned in results. Deeptools was used to performed the correlation between histone modifications with spearman model, which was added in “Materials and Methods” section. And it is indicated that only the H3K27me3 from PRO cells showed negative correlation with RNAPII and H3K27ac from DIF cells, and the H3K27me3 from DIF cells was negative correlated with RNAPII and H3K27ac from PRO cells.
15. Which analyses were performed on peak regions? and which on genome wide signal? More details should be added.
Response 15: The peaks of H3K4me3 H3K27ac and RNAPII were called with MACS2 using regular mode with default parameters, while broad mode was used as ‘macs2 callpeak -t ChIP.bam -c Control.bam -f BAMPE -g 2.7e+9 -n name --broad --broad-cutoff 0.05’ for peak calling of H3K27me3. And ChIPseeker were used to performed the peak region annotation with the gff3 file of the Sscrofa11.1 genome. The description of the peak analysis was added in ChIP-seq data analysis of “Materials and Methods” section.
16. Figure 4A –Is the color scheme relate to both genes and modifications? How the histone signal was summarised across DEGs in order to generate this plot? gene bodies? promoters?
Response 16: The color scheme was related to both genes and modifications in Figure 4A, and the histone signal was calculated through summarized the peak number of ± 3 kb TSS region of each DEG.
17. Please add Y axis label to figure S4A, and correct the legend (change the second (A) to (B))
Response 17: Added the Y axis label to figure S4A with log2(Fold change vs control). And legend was corrected.
18. Figure 5A is hard to interpret, it will be more informative to present the distribution of H3K27me3 peaks across genomic features using pie charts instead.
Response 18: Added pie charts to illustrate the distinction of the intersection sizes of H3K27me3 modification in supplementary Figure S5A, which was used to describe the details of Figure 5A, and legend was also added for Figure S5A.
19. Line 268: “Moreover, the upregulated 268 histone-modified DEGs were remarkable enriched in skeletal muscle tissue…” – The number of genes represented within these processes is between 2-4 and the enrichment is not highly significant. Taking this into consideration please use soft term instead of “remarkable enriched”, and also be careful in drawing conclusions regarding the role of H3K27me3 modification for gene activation during PSC differentiation from this observation.
Response 19: Indeed, as reviewer said, the term “remarkable enriched” would be replaced and the sentence was changed into “Moreover, the upregulated histone-modified DEGs were involved in skeletal muscle tissue…”
20. Line 273 – “MYOG and CDKN2B, were blocked with H3K27me3, of which the transcriptional activity was low at proliferation phase (Figure 5D) “ – while the figure definitely show increase in MYOG H3K4me3 labeling in DIF cells, no reduction in H3K27me3 is clearly seen. The authors should update the text and figure legend to describe observations that are supported by the data. Also please correct the legend “The IGV browser was used to show the ChIP-Seq and RNA-Seq enrichment” it should be “RNA pol II” instead of “RNA-seq”
Response 20: The results were updated. Added the landscape of CDKN2B and MYOG using IGV browser to supplementary Figure S5C and Figure S5D. The TSS region of MYOG was roomed in to illustrate the reduction of H3K27me3 during cell differentiation, which depleted especially at TSS region. Legend were modified for Figure S5C and Figure S5D. And legend were also corrected for Figure 5D.
21. List of DEGs should be given as supplementary table
Response 21: Added the list of 917 DEGs into supplementary as table S4.
22. Line 299: “Our results showed that it is the differential expression of these DEGs initiating cell differentiation to form myotubes, which were confirmed by qPCR.” – no experiments were done to test that the change in expression of the DEGs initiate cell differentiation. What the authors show is that these genes change their expression in differentiating cells. Please rephrase.
Response 22: The sentence would change into “Our results showed that these important DEGs were revealed differentially expressed in PSC differentiation, which were confirmed by qPCR.”
23. Line 318: “Our results emphasized the crucial roles of H3K27me3 modification during cell differentiation” – again, the results show that H3K27me3 modification dynamically changed during differentiation. In order to emphasize the crucial roles of H3K27me3 for differentiation, experiments should be done. Please rephrase.
Response 23: The sentence would change into “Our results emphasized the dramatic decrease of H3K27me3 modification during cell differentiation.”
24. English editing is required, please also change "multi-comb" to "polycomb" in line 56
Response 24: The word was corrected.

Reviewer 3 Report
In this manuscript, the authors analyzed the genome wide gene expression in porcine satellite cells. The authors claimed that the depletion of H3K27me3 promoted porcine satellite cell differentiation by regulating the expression of myogenic genes. Because H3K27me3 is involved in the downregulation of nearby genes, the depletion of H3K27me3 normally upregulates the gene expression. Although the authors used porcine model, the relation of H3K27Me3 and muscle differentiation was demonstrated previously (PNAS (2011), 108:E149-58, JCS(2013) 126:565-579). There are several points that should be addressed before publication.
1.Line 172. The gene numbers should be fixed.
2.In Figure 3A and Figure 4B. Control was used to compare the value, however there is no description for the control.
3. In Figure 3A. The authors should compare the enrichment value for H3K27me3 between PRO and DIF. These figures showed the similar pattern for H3K27me3. Instead, green lines appear to be more significant.
4. In Figure 3D. The heatmap data is for DIF sample. The authors should show the data for PRO sample. Is there any significant difference?
5. In Figure 2D, the authors should provide more detailed explanation for this graph. The authors should explain the upregulation of FoxO signaling terms.
6. In Figure 5D, there is no data for CDKN2B. The authors should fix the text, or provide the additional data for CDKN2B.
7. In Figure 5E, is there any quantitative data, which indicate the level of H3K27me3 is reduced in “DIF” condition?
Author Response
Reviewer 3:
In this manuscript, the authors analyzed the genome wide gene expression in porcine satellite cells. The authors claimed that the depletion of H3K27me3 promoted porcine satellite cell differentiation by regulating the expression of myogenic genes. Because H3K27me3 is involved in the downregulation of nearby genes, the depletion of H3K27me3 normally upregulates the gene expression. Although the authors used porcine model, the relation of H3K27Me3 and muscle differentiation was demonstrated previously (PNAS (2011), 108:E149-58, JCS(2013) 126:565-579). There are several points that should be addressed before publication.
1. Line 172. The gene numbers should be fixed.
Response 1: The number of detected genes was corrected.
2. In Figure 3A and Figure 4B. Control was used to compare the value, however there is no description for the control.
Response 2: The input files of our ChIP-seq data were regarded as control, which were used to normalize each ChIP-seq data and were mentioned in “Materials and Methods” section.
3. In Figure 3A. The authors should compare the enrichment value for H3K27me3 between PRO and DIF. These figures showed the similar pattern for H3K27me3. Instead, green lines appear to be more significant.
Response 3: The compare of the enrichment value of H3K27me3 between PRO and DIF was performed. Although the peak number of H3K27me3 decreased 50%, only 1626 peaks were detected depletion at TSS and genebody region, which showed similar pattern for H3K27me3 at TSS and genebody regions. While most of H3K27me3 peaks depleted at intergenic regions. And for RNAPII, it showed 3958 peaks decreased at TSS and genebody regions, which were significant changed in Figure 3A.
4. In Figure 3D. The heatmap data is for DIF sample. The authors should show the data for PRO sample. Is there any significant difference?
Response 4: The heatmap data of PRO was added into supplementary Figure S3A, which showed similar characteristic with different modification level compared with the result of DIF. Legend was added.
5. In Figure 2D, the authors should provide more detailed explanation for this graph. The authors should explain the upregulation of FoxO signaling terms.
Response 5: The description of Figure 2D were modified with more details to illustrate 5 top GO terms in second result. And for FoxO signaling term, 8 upregulated genes and 5 downregulated genes were revealed in DEGs, which tend upregulated with z-score larger than 0.
6. In Figure 5D, there is no data for CDKN2B. The authors should fix the text, or provide the additional data for CDKN2B.
Response 6: Added the landscape of CDKN2B using IGV browser to supplementary Figure S5C. Legend was modified.
7. In Figure 5E, is there any quantitative data, which indicate the level of H3K27me3 is reduced in “DIF” condition?
Response 7: In Figure 5E, we showed 6 red points at PRO phase which presented the H3K27me3 modification, while only 3 for DIF phase indicating the depletion of H3K27me3.

Round 2
Reviewer 3 Report
Although the authors responded extensively to the reviewer’s comment, there are several points that should be addressed before publication.
Previously, I commented like below.
“Because H3K27me3 is involved in the downregulation of nearby genes, the depletion of H3K27me3 normally upregulates the gene expression. Although the authors used porcine model, the relation of H3K27Me3 and muscle differentiation was demonstrated previously (PNAS (2011), 108:E149-58, JCS(2013) 126:565-579).”
And I asked the authors to explain the scientific advance and significance of the current manuscript, but the author did not respond.
In addition, I asked the authors to explain the evidences to conclude that the level of H3K27me3 is reduced in “DIF”, but the authors did not explain it seriously. They just counted the red spots in the figure.
“7. In Figure 5E, is there any quantitative data, which indicate the level of H3K27me3 is reduced in “DIF” condition?
Response 7: In Figure 5E, we showed 6 red points at PRO phase which presented the H3K27me3 modification, while only 3 for DIF phase indicating the depletion of H3K27me3.”
Author Response
Although the authors responded extensively to the reviewer’s comment, there are several points that should be addressed before publication.
Previously, I commented like below.
“Because H3K27me3 is involved in the downregulation of nearby genes, the depletion of H3K27me3 normally upregulates the gene expression. Although the authors used porcine model, the relation of H3K27Me3 and muscle differentiation was demonstrated previously (PNAS (2011), 108:E149-58, JCS(2013) 126:565-579).”
And I asked the authors to explain the scientific advance and significance of the current manuscript, but the author did not respond.
Response 1:
According to the scientific advance and significance, we’d like to explain three points: firstly, compared with using passage cell line like C2C12 cells, we primarily isolated the satellite cells from porcine skeletal muscle and the results will be more advantage to the research on myogenesis in pigs. Secondly, as adult stem cells in vitro, the satellite cells had a pluripotency including myogenesis, repair and adipogenesis et. al. which can further provide comprehensive information for the study on porcine trait improvements like intramuscular fat (IMF) and the muscle regeneration. Last, we adopted the genome-wide integrated analysis to combine epigenome and transcriptome to study the relationship between H3K27me3 and myogenesis and found the depletion of H3K27me3 related with 139 up-regulated DEGs, which will helpful for better understanding the myogenesis of pigs.
Compared with the similar previous studies (the reviewer mentioned PNAS 2011 and JCS 2013), our work was consistent with their results on H3K27me3 modification. Moreover, our results further demonstrated that not only myogenic DEGs, but also cell cycle related genes were regulated by histone modification in promoting myogenic differentiation such as CDKN2B. The integration analysis elaborately illustrated the change of the transcription profiles affected by H3K27me3 depletion. We added this to discussion section, and also cited these previous studies into the reference.
In addition, I asked the authors to explain the evidences to conclude that the level of H3K27me3 is reduced in “DIF”, but the authors did not explain it seriously. They just counted the red spots in the figure.
“7. In Figure 5E, is there any quantitative data, which indicate the level of H3K27me3 is reduced in “DIF” condition?
Response 7: In Figure 5E, we showed 6 red points at PRO phase which presented the H3K27me3 modification, while only 3 for DIF phase indicating the depletion of H3K27me3.”
Response 2:
We drew the conclusion mentioned by the reviewer mainly based on these evidences: Firstly, previous studies demonstrated that rapid reduction of H3K27me3 mark would occurred during specific stages of stem cell differentiation (Asp, P., et al., 2011; Agger, K., et al., 2007). Secondly, it was also observed during porcine muscle stem cell differentiation in this study, about half of the peak number of H3K27me3 modification was depleted from 22,060 to 11,573. While using ChIPseeker, intersection size of H3K27me3 modification region was illustrated that was dramatically decreased in intergenic, intron and promoter regions representing at Figure 5A and S5A. And quantitative data was also marked in the schematic diagram of Figure 5E and legend was corrected.
The other revised parts including: (1) revised Figure 5E and its legend; (2) revised the discussion part.
Round 3
Reviewer 3 Report
The authors' responses are adequate, and now the manuscript is acceptable.